# Discovering human diabetes-risk gene function with genetics and physiological assays

Heshan Peiris[1], Sangbin Park[1], Shreya Louis [1], Xueying Gu[1], Jonathan Y. Lam[1], Olof Asplund[2], Gregory C. Ippolito[3], Rita Bottino[4], Leif Groop [2], Haley Tucker [3] & Seung K. Kim[1,5,6]

Developing systems to identify the cell type-specific functions regulated by genes linked to type 2 diabetes (T2D) risk could transform our understanding of the genetic basis of this disease. However, in vivo systems for efficiently discovering T2D risk gene functions relevant to human cells are currently lacking. Here we describe powerful interdisciplinary approaches combining *Drosophila* genetics and physiology with human islet biology to address this fundamental gap in diabetes research. We identify *Drosophila* orthologs of T2D-risk genes that regulate insulin output. With human islets, we perform genetic studies and identify cognate human T2D-risk genes that regulate human beta cell function. Loss of BCL11A, a transcriptional regulator, in primary human islet cells leads to enhanced insulin secretion. Gene expression profiling reveals BCL11A-dependent regulation of multiple genes involved in insulin exocytosis. Thus, genetic and physiological systems described here advance the capacity to identify cell-specific T2D risk gene functions.

[1] Department of Developmental Biology, Stanford University School of Medicine, 279 Campus Drive, Stanford, CA 94305, USA. [2] Unit of Diabetes and Endocrinology, Lund University Diabetes Centre, Lund SE-205 02, Lund, Sweden. [3] Institute for Cellular and Molecular Biology and Department of Molecular Biosciences, University of Texas at Austin, Austin, TX 78712, USA. [4] Institute of Cellular Therapeutics, Allegheny Health Network, Pittsburgh, PA 15212, USA. [5] Department of Medicine (Endocrinology and Oncology Divisions), Stanford University School of Medicine, 279 Campus Drive, Stanford, CA 94305, USA. [6] Stanford Diabetes Research Center, Stanford University School of Medicine, 279 Campus Drive, Stanford, CA 94305, USA. Correspondence and requests for materials should be addressed to S.K.K. (email: seungkim@stanford.edu)

Type 2 diabetes (T2D) is a complex disease whose hallmarks include elevated circulating blood glucose levels. T2D risk is thought to arise from a combination of genetic and acquired factors, and intensive efforts in modern biomedical research are focused on understanding these. Findings from studies of T2D genetics, including genome-wide association studies (GWAS) conclusively demonstrate the polygenic basis of T2D risk.

GWAS studies have identified over 100 risk loci associated with T2D, impaired glucose control and insulin-related traits[1,2]. A prominent set of genes linked to T2D risk loci appears to function in regulating pancreatic islet beta cells, the sole source of systemic insulin in humans[1]. Thus identifying genetic factors which regulate insulin secretion by human beta cells is a crucial step to understand the genetics of T2D. However, the vast majority of T2D risk loci are associated with genes whose functions in glucose control and insulin output or responsiveness remain unknown. To cut through this Gordian knot in diabetes genetics, it would be useful to (1) use conditional genetics, (2) assess the effects of gene perturbation on relevant in vivo physiological phenotypes, like insulin output, and (3) discover ways to test findings in appropriate primary human cells, like islet beta cells in culture or in vivo.

Insulin function is conserved throughout the metazoan animal kingdom. Previously, we created systems in *Drosophila melanogaster* to assess in vivo physiology and genetics relevant to diabetes[3]. Fundamental features of metabolic and hormone regulation are conserved in humans and *Drosophila*[4]. Examples include: (1) production and secretion of systemic insulin by endocrine cells; islet beta cells in humans and insulin-producing cells (IPCs) in *Drosophila*[4,5]; (2) regulation of glucose metabolism by systemic levels of insulin, glucagon, leptins and incretins[5–7]; (3) conserved insulin signal transduction in orthologous insulin-target tissues[3,4]. In *Drosophila* IPCs and human beta cells, there is further remarkable conservation of the molecular and signaling mechanisms coupling glucose-sensing to insulin production and secretion[8]. Insulin-like peptide 2 (Ilp2) is the crucial insulin secreted by *Drosophila* IPCs for glucose regulation[9], and we have used an epitope-labeling strategy to measure circulating and total levels of Ilp2[10]. Powerful genomic-scale and cell biology resources developed in *Drosophila* have enabled genetic gain- or loss-of-function in IPCs or other cells to reveal that genetic regulation of insulin output in IPCs and human beta cells is also strikingly conserved.

Here we describe an experimental approach combining *Drosophila* genetics and insulin assays with human islet genetics to identify the function of genes linked to T2D risk. We identified *Drosophila* orthologs of human T2D-risk genes that regulate insulin output in *Drosophila*, which were not previously linked to insulin regulation in vivo. We used novel methods for genetic perturbation in primary human islets to test the function of T2D-risk genes in primary human islet cells. This identified a subset of genes like *BCL11A* that regulate human beta cell insulin secretion. Hence, our work provides an experimental paradigm for unraveling the complex knot of genetics underlying the pathogenesis of T2D.

## Results

**Using Drosophila to identify candidate T2D risk gene functions.** To assess T2D candidate risk gene function, we crossed gene-specific RNAi lines with tissue-specific driver lines to inactivate gene expression in *Drosophila* IPCs (functional orthologue of mammalian beta cells), and other insulin target tissues, like fat body (equivalent to vertebrate liver and fat). Ilp2 is the principal regulator of circulating glucose levels in *Drosophila*[11],

and prior studies established that nutrient and genetic regulation of Ilp2 production or secretion can be measured in adult *Drosophila*[10] using the epitope-labeled, biologically active analog, "Ilp2HF". Thus, measures of circulating and total Ilp2HF after RNAi permitted risk gene assessment. We then performed a loss-of-function screen of orthologues known to regulate human beta cell insulin output (Fig. 1a). For example, targeted inactivation in *Drosophila* IPCs of Ilp2HF (Fig. 1b), insulin receptor[10], or glucose transporter[10] (Glut1) severely reduced production or secretion of Ilp2, phenotypes also observed after loss of insulin, insulin receptor or glucose transporter in mammalian islet beta cells.

To verify further that our system accurately predicted gene function in regulating insulin secretion, we used RNAi in *Drosophila* IPCs to knockdown expression of genes encoding orthologs (see Methods) of six additional known human beta cell insulin regulators, including *GLIS3, ZNT8, ABCC8, DGKB, IGF2BP2* and *ADRA2* (Fig. 1b). In *Drosophila* IPCs, loss of *GLIS3, or ZNT8* orthologs impaired circulating Ilp2HF levels (Fig. 1b) and total insulin content (Supplementary Fig. 1a), similar to reduced insulin output observed after inactivation of these genes in mammalian islets[12,13]. By contrast, IPC knockdown of the *Drosophila* orthologs of *ABCC8, DGKB, IGF2BP2* or *ADRA2* increased circulating Ilp2HF levels in adults flies (Fig. 1b, Supplementary Fig. 1a), reminiscent of increased insulin output after genetic or pharmacological inactivation of these factors in pancreatic islets[14–16]. Thus, in 9/9 cases, targeted gene inactivation in IPCs led to changes of insulin output similar to that observed after analogous loss-of-function studies in pancreatic islets, providing striking evidence that genetic approaches in *Drosophila* could be useful for predicting the function of genes in human beta cells.

To discover roles of candidate diabetes risk genes in insulin regulation, we measured insulin output (total and circulating levels) after RNAi induced knockdown of T2D candidate risk factors, not known to be regulators of insulin production or secretion (Fig. 1c, Supplementary Fig. 1b). From an initial group of 40 candidate human diabetes risk genes (Fig. 1c, Supplementary Fig. 1b), we identified a subset of 14 genes (Fig. 1c) that had the following: (1) established expression in human islet cells, (2) an unambiguous *Drosophila* ortholog, and (3) an associated RNAi strain in *Drosophila* stock collections (see "Methods" section). From these 14 prioritized candidates, we identified 3 genes that significantly altered in vivo insulin output. Knockdown of *CG9650* or *fascetto*, orthologs of human *BCL11A* and *PRC1* respectively, enhanced circulating Ilp2HF levels. In the case of *CG9650* knockdown, this was associated with a slight increase in Ilp2HF content (Supplementary Fig. 1c). Knockdown of *optix*, an ortholog of human *SIX3*, significantly reduced circulating Ilp2HF levels (Fig. 1c). By contrast, circulating insulin levels were not detectably altered after knockdown of *CG9650* or *fascetto* in *Drosophila* fat body (Fig. 1d). To confirm the expression of *CG9650* in *Drosophila* IPCs, we visualized flies harboring a *piggyBac* transposon called PBacCG9650[CPTI001740], which encodes a GFP protein trap inserted within the CG9650 coding region[17]. We observed broad nuclear CG9650 expression within the brain, while the co-production of GFP and Ilp2 confirmed the expression of *CG9650* in *Drosophila* IPCs (Fig. 1e). We confirmed *CG9650* expression in *Drosophila* IPCs with a second independent *Drosophila* line (Supplementary Fig. 1d). The *CG9650* ortholog *BCL11A* encodes a transcriptional regulator of fetal hemoglobin switching[18] and other hematopoietic lineages, and the *fascetto* ortholog *PRC1* encodes a microtubule regulator of cell division and cytokinesis[19]. Both *BCL11A* and *PRC1* are expressed in human islet beta and alpha cells[20,21], but their roles in islet function have not been reported. SIX3 belongs to the *Sine Oculis* family of homeodomain transcription factors[22] and has

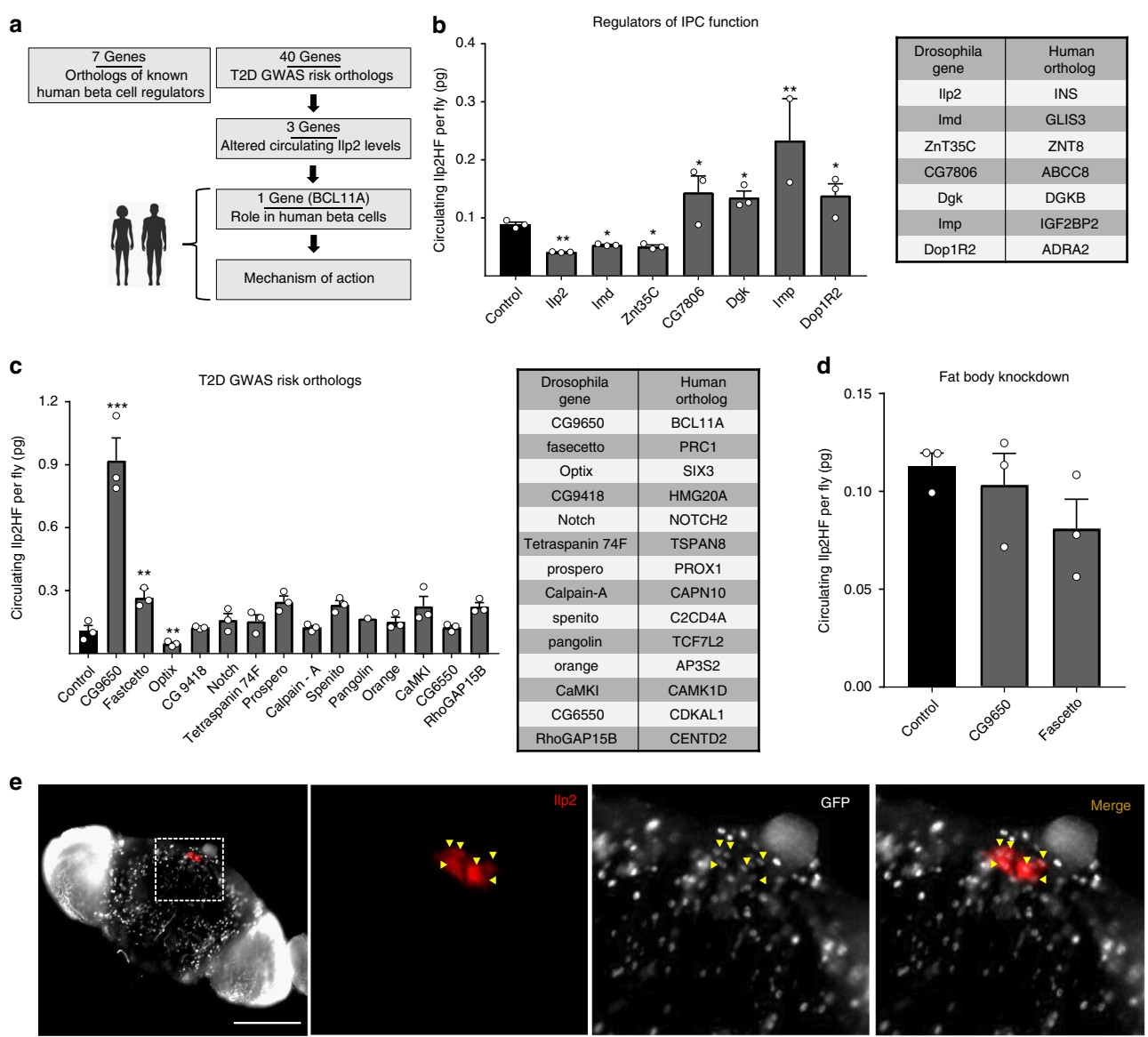

**Fig. 1** Screening T2D GWAS risk gene in Drosophila Melanogaster. **a** Schematic summary of this study. **b** Circulating Ilp2HF levels (picogram, pg) in ad libitum fed adult flies with IPC-specific RNAi knockdown of beta cell genes ($n = 3$ independent experiments, sampled in duplicate). **c** Circulating Ilp2HF levels (pg) in ad libitum fed adult flies with IPC-specific RNAi knockdown of GWAS candidate genes ($n = 3$ independent experiments, sampled in duplicate). **d** Circulating Ilp2HF levels (pg) in ad libitum fed adult flies with fat body-specific RNAi knockdown of CG9650, fascetto and Optix ($n = 3$ experiments, sampled in duplicate). **e** Immunostaining of adult Drosophila brains with antibodies recognizing GFP (white) or Ilp2 (red), scale bar = 100 μm. See also Supplementary Figure 1. The data presented as mean, error bars represent the standard error, and two-tailed $t$ tests were used to generate $p$ values. *$p < 0.05$, **$p < 0.01$, ***$p < 0.001$

age-dependent expression in human beta cells[20]. *SIX3* mis-expression in human islets can enhance insulin secretion[20], but the effects of *SIX3* loss in islets on insulin regulation are unknown. Thus, our screen identified multiple novel *Drosophila* regulators of in vivo insulin output that are orthologs of candidate human T2D risk genes, suggesting roles for these genes in human islet insulin regulation.

**Human islet *BCL11A* expression and impaired insulin secretion.** Previous studies have shown that human beta cell *BCL11A* expression declines significantly with age, suggesting that beta cell expression may be dynamically regulated by genetic and environmental factors[20,21]. To investigate this, we analyzed islet gene expression from subjects with T2D, or from non-diabetic controls[23]. Measures of in vitro islet stimulation index (ratio of

insulin secretion at high vs. low glucose: see "Methods" section) operationally identified three distinct groups: glucose responsive non-diabetics, glucose responsive T2D donors and non-responsive T2D donors (Fig. 2a). For further analysis, we selected 10 datasets from each group. We did not observe significant variation in the age, body mass index (BMI) or gender distribution among these groups (Supplementary Fig. 2a-c). However, the stimulation index of the isolated islets correlated well with blood HbA1c levels. The non-diabetic subjects with lower HbA1c levels (<5.7%) had a superior insulin stimulation index from isolated islets compared to non-responsive T2D subjects with higher HbA1c (>6.5%) who had islets with a lower stimulation index (Fig. 2b). *BCL11A* mRNA levels were significantly higher in islets from non-responsive T2D donors compared to non-diabetic donors (Fig. 2c). Thus, an increased level of islet *BCL11A*

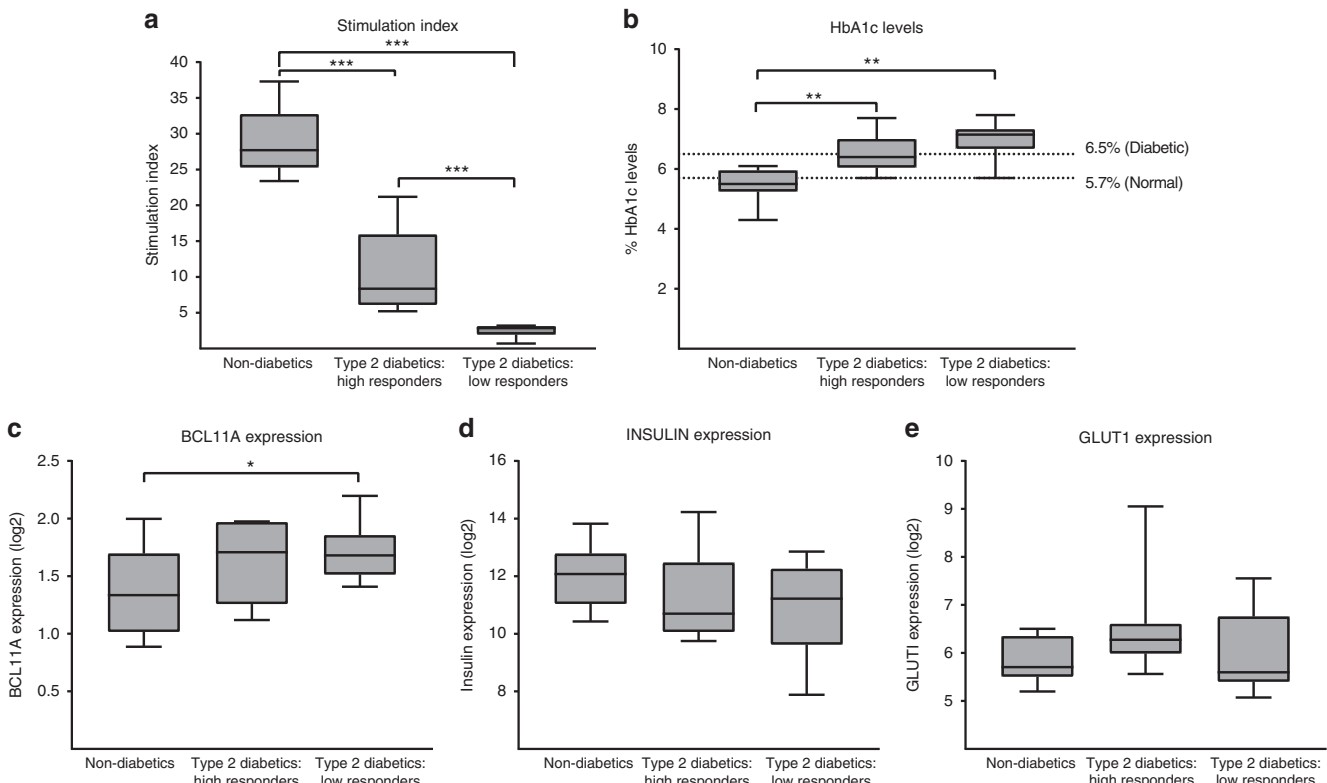

**Fig. 2** Gene expression changes in T2D human islets. **a** Stimulation index of isolated human islets in response to glucose among the three defined groups. **b** Donor Hba1c levels among the three groups, dotted lines indicate the levels of normal (below 5.7%), pre-diabetic (5.7–6.5%) and diabetic (above 6.5%) individuals. **c** BCL11A mRNA expression across the three groups. **d** INSULIN mRNA expression across the three groups. **e** GLUT1 mRNA expression across the three groups. See also Supplementary Figure 2. The data obtained from a total of 30 donors (10 per group), presented as boxplots, center line represents the mean, whiskers represent the minimum and maximum points, and two-tailed *t* tests were used to generate *p* values. *$p < 0.05$, **$p < 0.01$, ***$p < 0.001$

expression was linked to established diabetes and impaired insulin secretion. By contrast, mRNAs encoding INSULIN and GLUT1 were not significantly altered in these groups (Fig. 2d, e). Together, our findings suggest that *BCL11A* expression is elevated during T2D and chronic hyperglycemia. Additionally, we provide a strong, previously unknown link between increases in human islet BCL11A expression and decreased islet insulin secretion.

**BCL11A is glucose-responsive and inhibits insulin secretion.** Based on evidence that HbA1c elevation, indicating chronic hyperglycemia, was correlated with elevated islet *BCL11A* mRNA, we postulated that islet *BCL11A* expression might be regulated by glucose, an effect not previously reported. To test this, we cultured human islets ($n = 5$) from non-diabetic donors in 3 mM, 11 mM or 16 mM glucose for 5 days, then measured *BCL11A* mRNA levels by quantitative RT-PCR (qPCR, Donor Info: Supplementary Fig. 3a). Islets cultured at 16 mM glucose had a 50% increase in BCL11A expression compared to islets cultured at lower glucose concentrations (Fig. 3a). Together with our other findings, these data indicate that *BCL11A* expression is glucose-dependent. To test whether increased BCL11A is sufficient to inhibit human insulin secretion, we infected dispersed primary human islet cells (from non-diabetic donors) with lentivirus to overexpress-BCL11A (Fig. 3b), then re-aggregated cells into pseudoislets[20,24] to measure glucose-stimulated insulin secretion. Using immunohistology, we detected increased BCL11A production in beta cells after lentiviral-BCL11A overexpression compared to control pseudoislets expressing lentiviral GFP (Fig. 3c), a result corroborated by western blot (Fig. 3d) and

quantitative real-time PCR (qRT-PCR, Supplementary Fig. 3b). BCL11A overexpression did not alter insulin production and insulin secretion at 'baseline' glucose levels (2.8 mM), compared to controls (Fig. 3e). By contrast, BCL11A overexpression severely blunted insulin secretion after stimulation with glucose, or with glucose + IBMX (an established potentiator of insulin secretion). Moreover, we did not detect perturbation of pseudoislet glucagon secretion (Fig. 3f). Thus, our studies reveal that increased islet *BCL11A* expression is sufficient to inhibit islet insulin secretion.

**Identifying BCL11A target genes in human beta cells.** On the basis of known roles of BCL11A in transcriptional regulation, we postulated that gene expression profiling could suggest mechanisms by which BCL11A regulates insulin secretion from human beta cells (Fig. 4). For these studies, we reasoned that *BCL11A* gain-of-function induced by glucose or mis-expression might introduce non-specific changes, and instead attempted *BCL11A* loss-of-function. To achieve this, we used a novel approach combining lentiviral shRNA-based approaches with primary human pseudoislets to knockdown *BCL11A* expression (Fig. 4a, b, $n = 4$; Supplementary Fig. 4a). Lentiviral GFP co-expression permitted subsequent sorting and isolation of virally-infected cells. We optimized *BCL11A* knockdown after screening shRNAs targeting different regions of the *BCL11A* gene and selecting one that achieved 50% knockdown of *BCL11A* mRNA levels in islet cells (Supplementary Fig. 4b). To assess the effect of *BCL11A* knockdown specifically in beta or alpha cells, we used intracellular labeling and flow cytometry with insulin and glucagon antibodies[20] to purify human islet cells after infection with a

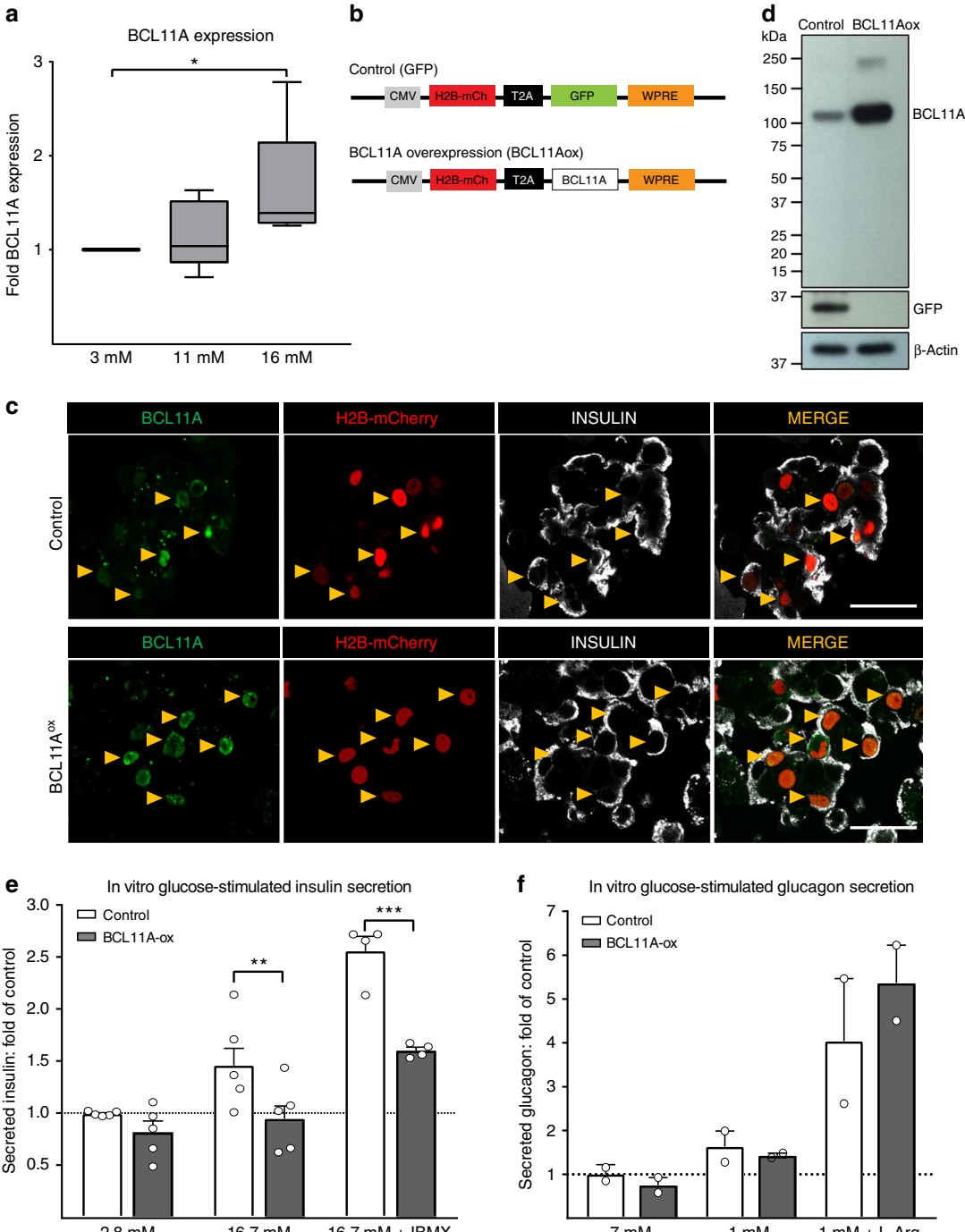

**Fig. 3** BCL11A expression in human islets is glucose responsive and inhibits insulin secretion. **a** BCL11A mRNA expression in human islets cultured at 3, 11 and 16 mM glucose for 5 days ($n = 5$). **b** Schematics of lentiviral construct used to overexpress BCL11A and control expressing GFP. **c** Immunohistochemical staining of human pseudo islets with BCL11A (green), Insulin (white) and H2B-mCherry (red) along with 4,6-diamidino-2-phenylindole (DAPI) nuclear staining (blue), scale bar = 20 μm. **d** Immuno blots showing the overexpression of BCL11A or GFP in relation to β-actin loading control ($n = 2$) in human pseudo islet lysates. **e** In vitro glucose-stimulated insulin secretion from human pseudo islets after transduction with control (white bars) or BCL11ox (black bars) lentiviral vectors. Secreted insulin is normalized to insulin content, ($n = 5$). **f** In vitro glucose-stimulated glucagon secretion from human pseudo islets after transduction with control (white bars) or BCL11ox (black bars) lentiviral vectors. Secreted glucagon is normalized to glucagon content, ($n = 2$). Original scan of blots are shown in Supplementary Figure 3. The data presented as mean, error bars represent the standard error, and two-tailed $t$ tests were used to generate $p$ values. *$p < 0.05$, **$p < 0.01$, ***$p < 0.001$

lentivirus expressing GFP (Fig. 4b, c). qRT-PCR confirmed the appropriate enrichment and depletion of each cell type in the purified GFP[+] cell fractions (Supplementary Fig. 4c), and also confirmed that human beta cells had over 50% reduction in *BCL11A* mRNA levels compared to controls (Fig. 4d: $n = 4$

independent experiments). Pearson correlation analysis followed by unsupervised hierarchical clustering of the sequenced libraries (Fig. 4e) revealed close clustering between control and BCL11A-kd samples from the same donor, reflecting the expected inter-donor variability we have previously reported[20]. We used

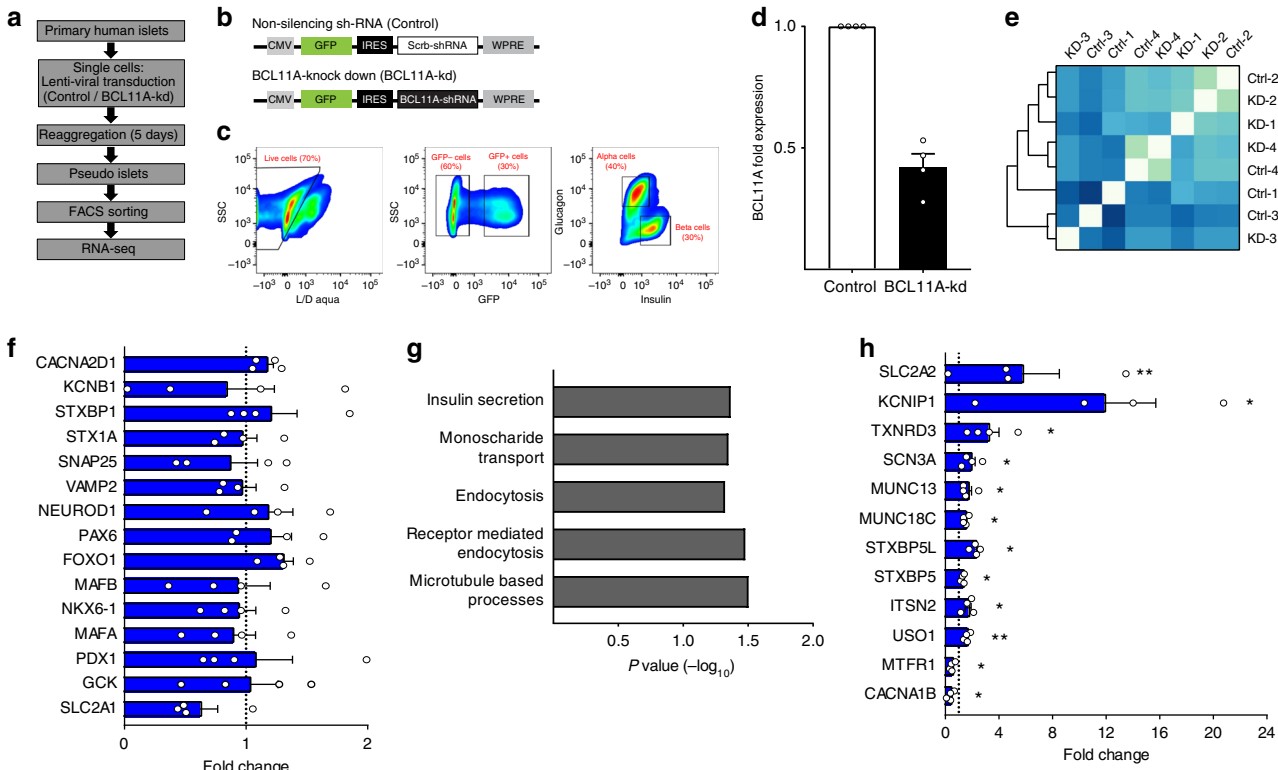

**Fig. 4** beta cell BCL11A regulates multiple pathways associated with human insulin secretion. **a** Workflow employed to generate RNA-seq libraries. **b** Schematics of lentiviral construct used to knockdown BCL11A expression. **c** FACS scheme used to obtain GFP expressing beta cells. **d** BCL11A mRNA expression in GFP expressing beta cells post FACS, control (white bar) or BCL11A-kd (black bar) ($n = 4$). **e** Pearson correlation coefficient matrix of all RNA-seq samples used in this study. **f** Fold transcript levels of genes critical for the maintenance of beta cell identity and function ($n = 4$). **g** GO terms enrichment in genes mis-regulated in beta cells post-BCL11A knockdown. **h** Fold transcript levels of genes significantly altered in beta cells post-BCL11A knockdown ($n = 4$). See also Supplementary Figure 4. The data presented as mean, error bars represent the standard error, and two-tailed $t$ tests were used to generate $p$ values. $*p < 0.05$, $**p < 0.01$

the DE-Seq algorithm to identify differentially expressed genes following *BCL11A* knockdown. This revealed 207 genes with significantly increased expression after *BCL11A* knockdown (Supplementary Table 1), while 203 genes were significantly decreased after knockdown (Supplementary Table 2). Thus, expression of the majority of genes expressed in human beta cells was unaltered after *BCL11A* knockdown, including multiple genes with known beta cell regulatory functions (Fig. 4f). Gene ontology (GO-term) analysis revealed that beta cells after *BCL11A* knockdown were enriched for expression of genes involved in pathways such as insulin secretion, endocytosis and microtubule processes (Fig. 4g). For example, *BCL11A* knockdown increased mRNAs encoding known enhancers of insulin secretion including MUNC18C and MUNC13, and regulators of secretory vesicle dynamics like ITSN2 and USO1[25,26] (Fig. 4h). MUNC18C is a pivotal member of the synaptic vesicle fusion pore complex, whose overexpression has been shown to improve vesicle dynamics, vesicle kinetics and neurotransmitter release[27]. MUNC13 is also a synaptic protein, and studies in insulinoma cell lines have previously demonstrated that increased MUNC13 expression significantly enhances insulin secretion[28]. Together, our combined approaches suggest that BCL11A is a crucial regulator of multiple genes critical for human beta cell insulin secretion and vesicle dynamics.

**Islet *BCL11A* loss stimulates insulin secretion**. On the basis of the observation that *BCL11A* knockdown increased the expression of known enhancers of insulin section, we postulated that

*BCL11A* knockdown might enhance human insulin secretion. To test this, we measured glucose-stimulated insulin secretion after knockdown of *BCL11A* expression in human pseudoislets using lentivirally-expressed shRNA ($n = 4$; Donor information Supplementary Fig. 5a). Knockdown of *BCL11A* was confirmed by qRT-PCR (Fig. 5a) following islet cell infection. Using in vitro glucose-stimulated insulin-secretion assays, we observed significant enhancement of insulin secretion at both basal (2.8 mM) and elevated (16.7 mM) glucose concentrations compared to controls (Fig. 5b). These data were normalized to total insulin content, which was not significantly altered after *BCL11A* knockdown. This suggests that enhanced insulin release from islets after *BCL11A* knockdown likely reflects improved secretion, consistent with our RNA-seq data after *BCL11A* knockdown (Fig. 4). To assess the durability of these responses after *BCL11A* loss, we transplanted pseudoislets in immunocompromised NSG mice[29] then measured glucose-stimulated insulin secretion 30 days after transplantation (see Methods). Following *BCL11A* knockdown, transplanted pseudoislets secreted more insulin before and after intraperitoneal glucose challenge (Fig. 5c). Human alpha cells also express *BCL11A*[20,21], but we did not observe altered glucagon secretion by pseudoislets after *BCL11A* knockdown (Fig. 5d).

To further investigate the effect of *BCL11A* on insulin secretion in vivo, we generated mice lacking *Bcl11a* expression in beta cells. We intercrossed Bcl11a^fl/fl mice with RIP-Cre mice[30] to generate RIP-Cre, Bcl11a^fl/fl progeny (hereafter, Bcl11aβKO mice) and littermate controls, including RIP-Cre, Bcl11a^fl:+ mice (hereafter, Control mice, see Methods). The lack of Bcl11a expression was

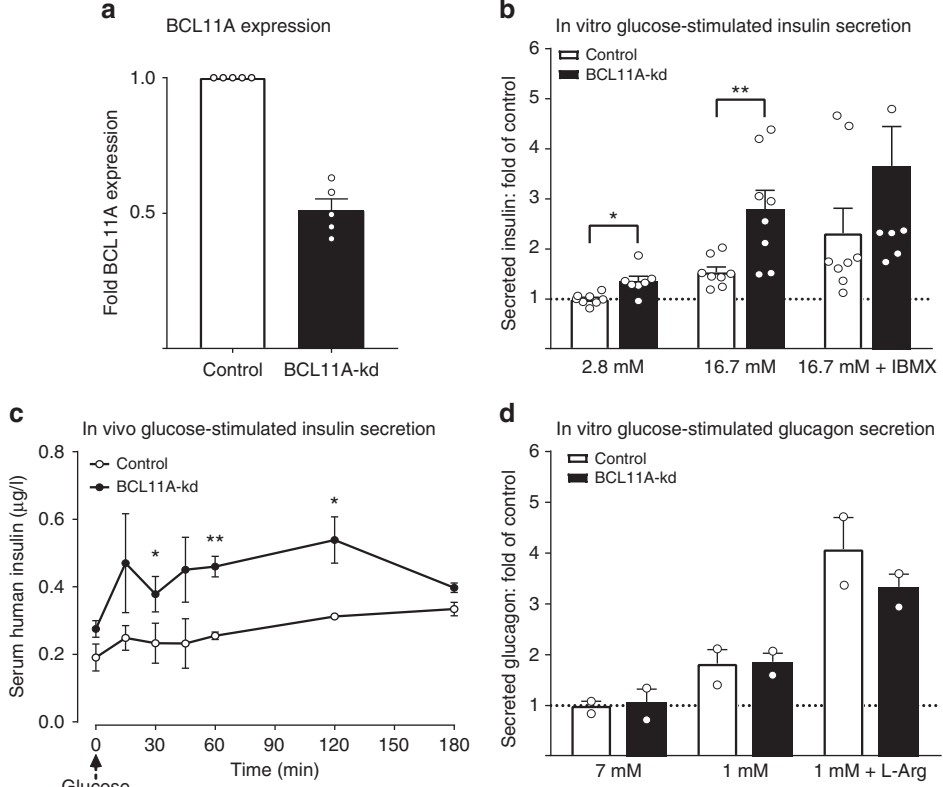

**Fig. 5** BCL11A knockdown enhances human insulin secretion. **a** BCL11A mRNA expression in human pseudo islets following transduction with control (white bar) or BCL11A-kd (black bar) lentiviral vectors ($n = 5$). **b** In vitro glucose-stimulated insulin secretion from human pseudo islets after transduction with control (white bars) or BCL11-kd (black bars) lentiviral vectors. Secreted insulin is normalized to insulin content, ($n = 5$, sampled in duplicate). **c** In vivo glucose-stimulated insulin secretion from human pseudo islets 30 days post-transplantation under the renal capsule of male NSG mice. Pseudo islets were transduced with control (white bars) or BCL11-kd (black bars) lentiviral vectors prior to transplantation. ($n = 3$). **d** In vitro glucose-stimulated glucagon secretion from human pseudo islets after transduction with control (white bars) or BCL11-kd (black bars) lentiviral vectors. Secreted glucagon is normalized to glucagon content, ($n = 2$). See also Supplementary Figures 5 & 6. The data presented as mean, error bars represent the standard error, and two-tailed $t$ tests were used to generate $p$ values. *$p < 0.05$, **$p < 0.01$

confirmed by immunohistological staining of pancreatic sections obtained from 12-week-old mice (Supplementary Fig. 5b, c). After glucose challenge of 8-week-old Bcl11aβKO and control mice, we did not observe significant differences in glucose clearance (Supplementary Fig. 5d). However, assessment of circulating insulin levels after glucose challenge revealed that Bcl11aβKO mice secreted significantly higher insulin levels for a sustained period (Supplementary Fig. 5e-f). To confirm that this enhancement of in vivo insulin secretion was not due to variations in insulin sensitivity, we isolated islets from 8-week-old Bcl11aβKO and control mice and performed in vitro perifusion assays (Supplementary Fig. 5g). Bc11aβKO islets secreted more insulin in response to high glucose concentration and beta cell depolarizing agents, confirming that the knockdown of Bcl11a enhanced insulin secretion similar to that observed in human pseudo islets. We previously showed that BCL11A knockdown in human pseudo islets resulted in a significant increase in GLUT2 expression in human beta cells (Fig. 4). To investigate if the downstream targets of BCL11A are conserved between human and mouse beta cells, we measured GLUT2 expression in Bcl11aβKO islets (Supplementary Fig. 6). We observed a significant increase in Glut2 expression in Bcl11aβKO islets compared to age-matched controls (Supplementary Fig. 6a-b), confirming that the mechanisms of BCL11A action and its downstream targets are conserved between human and mouse beta cells.

## Discussion

Elucidating the genetic and physiological basis of human diabetes involves understanding the function of multiple regulators in diverse tissues and organs in dynamic metabolic settings. Here we describe an integrated, multisystem approach incorporating *Drosophila* genetics and physiology, and human islet biology to identify in vivo tissue-specific functions of candidate T2D risk genes. These methods identified multiple T2D risk candidates as conserved regulators of insulin output in flies and humans. Focused physiological and genetic studies in humans and isolated human islets revealed BCL11A as an unsuspected regulator of islet beta cell gene regulation and function. We envision that these interdisciplinary approaches will be useful for systematically assessing tissue-specific functions of the rapidly growing set of diabetes candidate risk genes identified by GWAS and other modern genomic methods.

T2D is a disease involving multiple organs, including the pancreas, liver, skeletal muscle and fat; thus, in vivo systems that efficiently identify organ-specific functions of candidate diabetes genes are intensely sought. Prior findings highlight the growth and advantages of in vivo genetic and physiological methods in *Drosophila* for deconvoluting T2D genetics, including the striking conservation of signaling and genetic mechanisms regulating insulin output by *Drosophila* IPCs and human beta cells[3,5–7,10,31,32]. We used RNAi and tissue-specific driver lines to inactivate gene expression in *Drosophila* IPCs and fat body

(functional orthologs of islet beta cells, liver and adipose cells), and revealed nine *Drosophila* genes that regulate IPC insulin production and/or secretion (homologs of *GLIS3, ZNT8, ABCC8, DGKB, IGF2BP2, ADRA2, SIX3, PRC1* and *BCL11A*). Moreover, the direction of altered insulin output after genetic loss-of-function in IPCs in 9/9 cases matched those observed in analogous studies of pancreatic islet loss-of-function, in studies requiring less than 4 months. From 14 candidate T2D risk regulators, our fly-based screens identified three genes, *BCL11A, SIX3* and *PRC1*, as regulators of human beta cell function, and studies here and elsewhere (H.P., S.P., S.K., unpublished results) support this prediction. Thus, in contrast to other metazoan genetic systems like worms[33,34], zebrafish[35,36] and mice, or cultured mammalian cell lines[37,38], our genetic and physiological approaches with flies permitted screens of unprecedented scale, efficiency and predictive power that successfully prioritized and revealed in vivo tissue-specific functions of diabetes genetic regulators. Moreover, *Drosophila* additionally provides opportunities (not explored here) to assess gene function in fasting and re-feeding settings[10], and to target multiple genes in the same fly, with the potential to reconstitute polygenic diabetes risk seen in humans.

Systems to advance human islet genetic studies would accelerate investigations of T2D candidate risk gene function, a crucial step in deciphering diabetes genetics. Meta-analyses of GWAS have identified over 100 susceptibility loci associated with an increased incidence of T2D[1,38–41]. These include risk loci harboring *BCL11A, SIX3* and *PRC1*. Here we combined pseudoislet development[20,24] with shRNA-based knockdown, islet transplantation and insulin measures to assess *BCL11A* function. This gene was identified as a potential novel beta cell regulator by our *Drosophila* studies, but its function in islets was not previously reported. *BCL11A* encodes a zinc-finger transcription factor whose expression is dynamically regulated in blood cells, and whose targets include fetal globin genes[42–47]. A similar dynamic expression pattern of *BCL11A* has previously been observed in human beta cells[20,21] with higher *BCL11A* expression in juvenile beta cells and diminished *BCL11A* expression in adult beta cells. Remarkably, this decline of *BCL11A* expression corresponds with age-dependent increases of basal and stimulated insulin secretion in human islets[20]. Multiple T2D GWAS SNPs map to the BCL11A locus[40,41], but it remains unknown if these polymorphisms affect beta cell *BCL11A* expression. Recent single cell RNA-Seq studies suggest that BCL11A expression in islets from T2D subjects is increased in beta cells but not alpha cells[48]. Based on these findings and the results reported here, we speculate that some T2D GWAS risk SNPs might increase beta cell *BCL11A* levels and negatively affect glucose-stimulated insulin secretion. However, additional beta cell-specific eQTL studies of *BCL11A* and SNP meta-analysis combined with chromatin conformation studies[49,50] are needed to test this possibility. Sub-chronic exposure to elevated glucose levels stimulated *BCL11A* expression in primary human islets from cadaveric donors without diabetes (Fig. 3a), providing evidence that external cues could also influence islet *BCL11A* levels. Supporting this observation, we found that *BCL11A* mRNA in T2D islets exposed in vivo to chronically-elevated glucose was significantly higher, and correlated with reduced insulin secretion. These findings raise the likelihood that potential islet *cis*-regulatory elements linked to *BCL11A* will need assessment in both physiological and pathophysiological settings, including normoglycemia and hyperglycemia.

Our findings provide strong evidence that *BCL11A* is a potent suppressor of insulin secretion and could have dynamic roles in T2D pathogenesis. We show that *BCL11A* expression in human islets is dynamically increased by high glucose and that similar

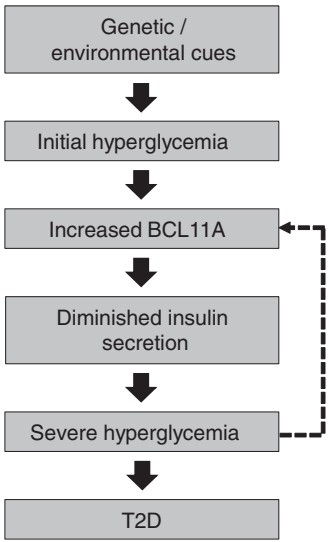

**Fig. 6** Role of BCL11A in T2D pathogenesis and beta cell function. Schematic diagram describing the role of BCL11A in human beta cell dysfuction and the pathogensis of T2D

increases in islet BCL11A levels are sufficient to inhibit insulin secretion by human beta cells. We suggest that hyperglycemia may initiate a "feed forward" sequence of elevated *BCL11A* expression leading to impaired insulin secretion and worsening hyperglycemia, like that seen in T2D patients (Fig. 6). However, inappropriately reduced *BCL11A* expression could also result in a pathological sequence of chronic, relatively increased insulin secretion, leading to insulin resistance, beta cell exhaustion, and eventual hyperglycemia. This sequence is observed in mice expressing multiple copies of the insulin gene[51] and in patients with primary hyperinsulinemia, like those receiving chronic insulin infusion[52] or subjects with insulinomas. Thus, our work reveals multiple features of islet BCL11A regulation and function, including (1) glucose-dependent islet *BCL11A* expression, (2) evidence that dynamic changes of *BCL11A* expression can reduce or increase insulin secretion, and (3) identification of BCL11A target genes governing insulin trafficking and secretion. Future investigations of *SIX3* and *PRC1*, two additional T2D risk gene candidates nominated as regulators of insulin output by our *Drosophila* screen, could identify whether these genes also regulate human islet insulin secretion.

In summary, studies here reveal that approaches combining genomic-scale genetics and physiology in flies, together with GWAS and physiological studies in humans, and human islet genetics, provide a powerful integrated system to assess cell-specific gene function and to clarify the genetic basis of human islet beta cell dysfunction in diabetes.

## Methods
**Drosophila Studies**. Drosophila orthologs of human genes were identified via the comparative genomics section of the Ensembl project. TRiP RNAi lines were obtained from Bloomington Drosophila Stock Center. To knockdown genes in adult IPCs, TRiP RNAi lines were crossed to the UAS-Dcr-2.D; Ilp21 gd2HF(attP2) Ilp215–1-GAL4 strain. To knockdown genes in adult fat body, TRiP RNAi lines were crossed to the UAS-Dcr-2.D; Ilp21 gd2HF(attP2) Lk6-GAL4 strain. To measure circulating and total Ilp2HF content per fly, Ilp2HF ELISA[10] was used with the following modifications: fifteen ad libitum fed 3-day-old male flies abdomens were dissected open, submerged in 50 µl of PBS with 0.2% Tween 20, and vortexed for 30 min at room temperature. The supernatants of hemolymph extraction were transferred to anti-FLAG antibody coated ELISA plate wells containing 50 µl of anti-HA-peroxidase at 2.5 ng/ml in PBS with Tween 20. Two CPTI protein trap lines for CG9650, CPTI000886 and CPTI001740, were obtained from Kyoto Stock Center. Immunostaining for GFP and Ilp2 protein in adult brains was performed as previously described[10] with the following modifications: mouse

anti-GFP antibody (1:1000; Invitrogen), rabbit anti-Dilp2 antibody (1:1000) (7), Alexa Fluor 488 and 647 secondary antibodies (1:1000; Invitrogen) were diluted and incubated in PBS with 0.3% Triton-X100.

**Human Islet Procurement**. De-identified human islets were obtained from healthy, non-diabetic organ donors with less than 15 h of cold ischemia time, and deceased due to acute trauma or anoxia. Organs and islets were procured through the Integrated Islet Distribution Network (IIDP) and the Alberta Diabetes Institute Islet Core.

**RNA Extraction and quantitative RT-PCR**. RNA was isolated from human islets or pseudo islets using the PicoPure RNA Isolation Kit (Life Technologies). cDNA was synthesized using the Maxima First Strand cDNA synthesis kit (Thermo Scientific) and gene expression was assessed by PCR using the Taqman Gene Expression Mix (Thermo Scientific). The following Taqman probes (Life Technologies) were used; ACTIN-B, Hs4352667_m1; BCL11A, Hs01093197_m1; INSULIN, Hs00355773_m1; GLUCAGON, Hs00174967_m1; CPA-1, Hs00156992_m1.

**Lenti-virus production**. Lenti-viruses were produced by transient transfection of HEK293T cells with pCDH lentiviral backbone vectors and pMD2.G (12259; Addgene) and psPAX2 (12260; Addgene) packaging constructs. Turbofect reagents were used for transfection (Life Technologies). Supernatants were collected and purified using PEG-it (System Biosciences). Concentrated lenti-virus was stored at 80 °C for transduction of primary human cells.

**Human pseudo islet generation**. Human islets were dissociated into a single cell suspension by enzymatic digestion (Accumax, Invitrogen). For each experimental condition, ~1 × 10⁶ cells were transduced with lenti-virus corresponding to 1 × 10⁹ viral units in 1 ml as determined by the Lenti-X qRT-PCR titration kit (Clonetech). Lenti-viral transduced islets cells were cultured in 96-well ultra-low attachment plates (Corning) and cultured for 4 days at 37 °C in 5% $CO_2$. After 4 days, pseudo islets were transferred to a 6 well plate and cultured overnight prior to further molecular or physiological analysis. Culture media: RPMI 1640 (Gibco), 2.25 g/dl glucose, 1% penicillin/streptomycin (v/v, Gibco) and 10% fetal bovine serum (HyClone).

**Immunohistochemistry**. Human pseudo islets were fixed for 1 h at 4 °C and embedded in collagen (Wako Chemicals), mouse pancreata were fixed in 4% paraformaldehyde overnight at 4 °C. Ten μm-thick frozen sections were cut and stained following standard cryostaining protocols. Briefly, sections were washed in PBS, incubated with blocking solution (Vector Labs, SP-2002) at room temperature, followed by incubation in permeabilization/blocking buffer (1% bovine serum albumin, 0.2% non-fat milk, 0.5% Triton-X in PBS) for 1 h. Primary antibodies were mixed with permeabilization/blocking buffer at appropriate concentrations and slides were incubated at 4 °C overnight. The following primary antibodies were used: Guinea pig anti-Insulin, (1:1000, DAKO, A0564), rabbit anti-BCL11A (1:1000, Bethyl Laboratories, A300–380), mouse anti-mCherry (1:500, Abcam, ab125096), goat anti-Glut2 (1:300, Santa Cruz Biotechnology, sc-7580). Slides were washed with PBS, incubated with secondary antibodies at room temperature for 2 h. Following the final wash with PBS, slides were preserved with mounting medium containing DAPI (Vector Labs, Vectashield H-1200). Images were obtained using a Leica SP2 confocal microscope.

**Western blot**. Protein from approximately 500 human pseudo islets was extracted by sonication in RIPA buffer (Thermo Scientific) supplemented with a Protease inhibitor cocktail (Thermo Scientific). Samples were denatured and resolved on a 4–15% gradient precast protein gel (Biorad). Proteins were transferred onto a PVDF membrane (Biorad) and blocked with 5% non-fat dry milk (Thermo Scientific). Membranes were probed for a series of proteins with the following antibodies: rabbit anti-BCL11A (1:1000, Bethyl Laboratories, A300–380), HRP-conjugated goat anti-Rabbit secondary antibody (1:7500, Santa Cruz Biotechnology, sc-2004), rabbit anti-GFP (1:12,000, Life Technologies, A1112) and HRP-conjugated mouse anti-Actin (1:15,000, Abcam, ab49900). Protein abundance was visualized via the SuperSignal West Pico PLUS Chemiluminescent Substrate kit (Thermo Scientific).

**In vitro insulin and glucagon secretion assays**. Batches of 25 pseudo islets were used for in vitro secretion assays. Pseudo islets were incubated at a glucose concentration of 2.8 mM for 60 min as an initial equilibration period. Subsequently, pseudo islets were incubated at 2.8 mM, 16.7 mM and 16.7 mM + IBMX glucose concentrations for 60 min each. Pseudo islets were then lysed in an acid-ethanol solution to extract the total cellular insulin or glucagon content. Secreted human insulin or glucagon in the supernatants and pseudo islet lysates were quantified using either a human insulin ELISA kit or glucagon ELISA kit (both from Mercodia). Secreted insulin levels was divided by total insulin content and presented as a percentage of total insulin content, a similar method of data analysis was employed for glucagon secretion assays. All secretion assay were carried out in

RPMI 1640 (Gibco) supplemented with 2% fetal bovine serum (HyClone) and the above mentioned glucose concentrations.

**Intracellular staining and FACS sorting of human islet cells**. Pseudo islets were dispersed into single cells by brief enzymatic digestion (Accumax, Invitrogen). Cell were stained with the fixable viability dye Aqua Dead Cell Stain (Life Technologies). Cells were then fixed with 4% paraformaldehyde and permeabilized with Cell Perm Buffer (BioLegend). Subsequently cells were stained with primary and secondary antibodies. All incubations were carried out at 4 °C for a period of 30 min. The following antibodies were used for FACS experiments described in this study: Guinea pig anti-insulin (1:200, Dako), mouse anti-glucagon (1:200, Sigma), donkey anti-guinea pig Alexa Fluor-555 (1:250, Invitrogen), donkey anti-mouse Alexa Fluor-647 (1:250, Invitrogen). Labeled cells were sorted on a special order 5-laser FACS Aria II (BD Biosciences) using a 100 μm nozzle, with appropriate compensation controls and doublet removal. Sorted cells were collected into low retention tubes containing 50 μL of FACS buffer (2% v/v fetal bovine serum in phosphate buffered saline) supplemented with Ribolock RNase inhibitor (Thermo Scientific). Cytometry data was analyzed and graphed using FlowJo software (TreeStar v.10.8).

**RNA isolation and preparation of RNA-seq libraries**. A total of 5000 sorted, fixed beta cells were used for RNA-seq library construction. Cells were lysed in 100 uL of extraction buffer with 4 μL of protease inhibitor (RecoverALL isolation kit, Invitrogen by Thermo Fisher Scientific). Total RNA was isolated following the manufacturer's protocol with the on-column DNAse 1 treatment. RNA quality was assessed using Bioanalyzer RNA Eukaryote Pico chip (Agilent Technologies). SMART-Seq v4 Ultra Low input RNA kit (Clontech) was used to generate amplified cDNA which was subsequently sheared resulting in 200–500 bp fragments. RNA-seq libraries was generated using the Low Input Library Prep Kit v2 (Clontech) as per the manufactures instructions. Barcoded libraries were then multiplexed and sequenced as paired-end 150 bp reads on the Illumina HiSeq4000 platform.

**Bioinformatic and statistical analysis of RNA-seq data**. Each condition experimental had four biological replicates WT1-WT4 and KD1-KD4. Each data set had an average of 79.3 million x 126 bp long paired-end reads. Fastqc (version 0.11.4) was used to assess the quality of each sequencing run. Reads were then aligned to the human (hg19) using Bowtie version 2.2.7 with splice junctions being defined in GTF file (obtained from UCSC). An average of 73.6% of reads from all samples were aligned to the reference transcriptome. Expression at gene level was determined by calculating reads per kilobase per millions aligned reads (FPKM) as well as raw count using RSEM (version 1.2.30). Differentially expressed genes with fold change were further detected by DEseq2 version 1.10.1 for the two experimental conditions. GSEA (version 2.2.0) was used for gene set enrichment analysis.

**Transplantation and in vivo assessment of pseudo islet function**. Human pseudo islets were transduced and cultured as per the usual protocol previously described. Batches of 250 pseudo islets were resuspended in cold Matrigel and transferred into the left renal capsular space of host animals using a glass micro-capillary tube. Transplant recipients were 8-week-old male NOD scid IL2Rγ^null mice (stock number 005557; The Jackson Laboratory) and were anesthetized using ketamine/xylazine. Appropriate depth of anesthesia was confirmed by lack of toe-pinch response. 1 month post transplantation, mice were administered an intra-peritoneal glucose injection at a dosage of 3 g kg-1 body weight. Blood samples were collected via the tail vein at 0, 15, 30, 45, 60, 120 and 180 min post glucose injection. Serum human insulin levels were measured by a human insulin ELISA kit (Mercodia).

**Generation of Beta cell-specific Bcl11a knockout mice**. Bcl11a female mice homozygous for the floxed Bcl11a allele (Bcl11a^fl/fl)[43] were mated with Bcl11a^fl/+: RIP-Cre males. Five week old male Bcl11a^fl/+: RIP-Cre mice were used as controls, while age-matched Bcl11a^fl:fl:RIP-Cre mice were used as beta cell-specific Bcl11a knockout mice (Bcl11aβKO). Intraperitoneal glucose tolerance tests were carried out by injecting glucose at a dosage of 2 g per kg body weight, following a 6 h period of fasting. Circulating blood glucose and serum insulin levels were measured at 0, 5, 15, 30, 45, 60, 120 and 180 min post glucose injection via tail bleed, using a Contour glucometer (Bayer) and an ultra-sensitive mouse insulin ELISA kit (Mercodia).

**Study Approval**. Human islet studies were performed on islets of de-identified donors after informed consent, and approval by the ethics committee at Lund University and the Stanford University Institutional Review Board (IRB). All animal experiments were approved by and performed in accordance with the guidelines provided by the Stanford University Institutional Animal Care and Use Committee (IACUC).

## Data availability

All sequencing data that support the findings of this study have been deposited in the National Center for Biotechnology Information Gene Expression Omnibus (GEO) and are accessible through the GEO Series accession number GSE116369. All other relevant data are available from the corresponding author on request.

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

## Acknowledgements

We gratefully acknowledge organ donors and their families for tissue procurement. We thank Professors Mark McCarthy and Anna Gloyn, and Dr. Anubha Mahajan (University of Oxford) for advice on SNP analysis and data presentation; Drs. Efsun Arda and James Lee for protocols and help with intracellular sorting, and members of the Kim lab and K. Mackenzie for comments on the manuscript. H.P was supported by postdoctoral fellowships from Stanford Child Health Research Institute (UL1 TR001085) and the American Diabetes Association (1–16-PDF-086). Work in H.T's lab was supported by the National Institute of

Health (grant number F32CA110624 and R01CA31534), Cancer Prevention Research Institute of Texas (grant number RP120459) and the Marie Betzner Morrow Centennial Endowment. Work in the Kim lab was supported by the NIH (DK104211, DK107507, DK102612, P30 DK116074 and DK108817), an opportunity pool award through UO1DK105554 (to Dr. J. Florez, Broad Inst.) in the AMP T2D consortium, Juvenile Diabetes Research Foundation, the Leona M. and Harry B. Helmsley Charitable Trust, the H. L. Snyder Medical Foundation, and the Islet Research Core of the Stanford Diabetes Research Center.

## Author Contributions

Conceptualization: H.P., S.P., S.K.K.; Methodology: H.P., S.P., S.K.K.; Software: H.P., S.L.; Validation: H.P., S.P., S.L., S.K.K., Formal Analysis: H.P., S.P., S.L.; Investigation: H.P., S.P., S.L., X.G., J.Y.L., O.A., G.C.I., H.T., L.G., S.K.K.; Resources: W.A., R.B., H.T., S.K.K.; Data Curation: H.P.; Writing-Original Draft: H.P., S.K.K.; Writing-Review & Editing: H.P., S.P., G.C.I., H.T., L.G., S.K.K.; Visualization: H.P.; Supervision: H.T., L.G., S.K.K.; Project Administration: S.K.K.; Funding acquisition: S.K.K.

## Additional information

**Competing interests:** The authors declare no competing interests.

