## [Peer Review File · Nature Communications]

Response to Referees' Comments

Reviewer #1 (expert in Drosophila metabolism, RNAi screening) (Remarks to the Author):

This is a very elegant study, combining the power of Drosophila genetics with work in mammalian systems to identify a novel regulator of insulin secretion in islet cells. The data are of good quality, and the findings are exciting. This study will be of interest to a broad audience of people studying diabetes, insulin signaling, and metabolic control. I have only minor comments for improvement of the manuscript:

We thank the reviewer for his/her positive comments and enthusiasm.

Minor Issues:

1. *Since this paper will likely be of interest to non-drosophila biologists, it would be helpful to briefly introduce fly-specific terms and tissues at their first instance (e.g. fat body or driver line).*

We thank the reviewer for this suggestion. We have revised the text on page 5 to provide a better description of driver lines, Insulin producing cells and fatbody.

2. *Page 5 "For example, targeted inactivation in Drosophila insulin-producing cells (IPCs) of *Ilp2HF* itself (Figure 1b), insulin receptor, or glucose transporter (*Glut1*) severely reduced production or secretion of *Ilp2*, phenotypes also observed after loss of insulin, insulin receptor or glucose transporter in mammalian islet beta cells."*

*It is not clear where the data for *InR* or *Glut1* knockdown are? If not shown, it should be referenced as such to prevent the reader from looking around and not finding them.*

We thank the reviewer for pointing this out. We have previously published the effects of *InR* or *Glut1* knockdown on *Ilp2HF* production and secretion (Park S, 2014: Reference number 10 in this manuscript). This reference has been incorporated into the text highlighted by the reviewer on page 5 of the revised manuscript.

3. *Fig 1b: I don't have internet access right now, but I think "*Imp*" is not the official Flybase gene symbol for this gene? If not, the official gene symbol should be used (also for all other genes in all figures).*

We thank the reviewer for pointing this out. *Imp* is the standard gene symbol for IGF-II mRNA-binding protein gene and a fly ortholog of human IGF2BP2 (<http://flybase.org/reports/FBgn0285926>). This has been changed in revised Figure 1b and on pages 5 of the revised manuscript. All the other genes listed in Figure 1b and Supplementary Figure 1b have also been checked for consistency.

4. *Page 6 - when mentioning the co-expression of *CG9650-GFP* with *Ilp2* in the IPCs, it should also be noted in the text that the expression of *CG9650* is quite broad in the brain.*

We thank the reviewer for this suggestion and have modified the text on page 6 to also mention the broad expression of *CG9650* within the Drosophila brain.

5. *Fig 1e: Is *CG9650-GFP* localized in nuclei?*

We thank the reviewer for this question. We consistently observe nuclear expression of GFP in the PBacCG9650^{CPTI001740} and PBacCG9650^{CPTI000886} lines. As outlined in Supplementary Figure 1c the GFP protein trap is inserted within the coding region of

CG9650. Hence, the nuclear GFP staining is representative of the nuclear expression of the transcription factor CG9650. The text on page 6 of the revised manuscript has been modified to make mention of the nuclear expression GFP/CG9650.

6. Page 7: "Together, our findings suggest that T2D and chronic hyperglycemia could increase islet BCL11A expression, and that BCL11A suppresses islet insulin secretion." This sentence at this point in the manuscript is too speculative based on the data presented up to that point. It should be rephrased or removed.

We thank the reviewer for this suggestion and have rephrased this sentence to better reflect the data presented up to that point (page 7-8 of the revised manuscript).

7. Fig 4g - both the figure legend ("GO terms enriched in beta cells post-BCL11A knockdown") and the corresponding text don't make much sense (GO terms are not enriched in cells, but rather in lists of genes). I assume the authors mean that this figure is showing GO-term enrichment in the group of genes that are up-regulated upon BCL11A knockdown? (Or in the group of genes that is mis-regulated, both up and down?).

We thank the reviewer for pointing this out. The figure legend associated with Figure 4g has been changed and now reads 'GO terms enrichment in genes mis-regulated in beta cells post-BCL11A knockdown'. The text associated with this figure on page 9 has also been changed.

8. Fig 4h - I don't see MUNC13 and MUNC18 in the figure, as stated in the text? (I guess MUNC13 is perhaps the same as UNC13B? But I don't see MUNC18 at all?).

We regret this discrepancy in nomenclature and have made the following changes to Figure 4h: UNC13B replaced by MUNC13 and STXBP3 replaced by the more commonly used MUNC18C. Within the manuscript we now refer to these genes as MUNC13 and MUNC18C respectively. We thank the reviewer for this suggestion.

9. There seems to be something wrong with Fig 5d because the values, according to the y-axis, are "fold of control", but the control bar (white) is not equal to 1? Also, the drop in glucagon secretion upon BCL11A knockdown seems significant (the error bars are not overlapping, which would be fine), unlike what is stated in the text?

We thank the reviewer for pointing this out. To better assess the effects of BCL11A knockdown on glucagon secretion we have repeated this experiment with islets from two additional donors. As requested, we have assessed glucagon secretion in response to two glucose concentrations (7mM and 1mM) and the glucagon secretagogue L-arginine in pseudo islets (new Figure 5d). The data confirms that BCL11A knockdown did not affect glucagon secretion from human alpha cells. If the reviewer or editors require experiments with pseudo islets from a third donor, we are happy to do so.

10. Suppl. Fig. 5c - here too the error bars are not overlapping, and hence there seems to be a mild but statistically significant impairment in glucose clearance in the BCL11AbKO mice, despite elevated serum insulin, unlike what is stated in the text? If this is the case, the text should be fixed. Furthermore, there might be something interesting going on (e.g. the KO might be inducing peripheral insulin resistance due to persistently elevated circulating insulin levels?).

We thank the reviewer for raising this important point. The data presented in Supp Figure 5c (Supp Figure 5d in this resubmission) indicates that there is no statistical significance in glucose clearance between the Control and Bcl11a β KO mice.

Reviewer #2 (expert in beta cell biology) (Remarks to the Author):

In this article the authors take advantage of the Drosophila genetics to identify Drosophila orthologs of T2D risk genes that are regulators of insulin production in human islets. The authors identified BCL11A, a transcription factor, and showed that its loss of function leads to increased insulin secretion. BCL11A expression was found to be increased in T2D islets with impaired insulin secretion. RNA-seq data from human beta cells that has 50% reduction in BCL11A levels indicated altered regulation of genes involved in insulin exocytosis. This is an interesting and novel study and a couple of additional pieces of information could potentially strengthen the manuscript:

We thank the reviewer for this positive assessment.

1. Figure 3d. Control islets do not show any endogenous expression of BCL11A. Authors should comment on that.

We thank the reviewer for pointing this out. To confirm the endogenous expression of BCL11A in control islets we performed additional immunoblots with new chemiluminescent reactions capable detecting proteins of low abundance. As shown in new Figure 3d we are now able to detect the expression of BCL11A in control pseudo islet lysates as well as the expected higher BCL11A expression in the BCL11Aox pseudo islet lysates. The membrane was also probed with antibodies against GFP (present only in control samples) and beta-actin (loading control). We feel the manuscript is improved by this point, thank you.

2. Figure 3f, 5d: The effect of loss or overexpression of BCL11A on alpha cell function is not clear. Glucagon production upon serum starvation or utilization of glucagon secretagogues might be relevant to assess this.

We thank the reviewer for this suggestion. To assess the effects of BCL11A loss or overexpression on alpha cell function we have measured glucagon secretion in response to two glucose concentrations (7mM and 1mM) and the glucagon secretagogue L-arginine in pseudo islets from two additional human donors, using multiple technical replicates. This data is presented in new Figures 3f and 5d, and confirms that neither the loss nor overexpression of BCL11A affected glucagon secretion by alpha cells. We are confident in the quality and conclusions from these data. However, if the reviewer or editors require experiments with pseudo islets from a third donor, we are happy to do so.

3. It is not clear why authors chose to use pseudo islets after transduction. During the reaggregation, other islet cells transduced with the construct might influence beta cell gene expression pattern. Sorting the beta cells after transduction and performing RNAseq could have given more robust data. Authors should discuss and address this.

We regret that we were apparently unclear about the scheme of this experiment. We performed our RNA-seq analysis on populations of purified beta cells. As outlined in Figure 4c, following the gating for viable cells, we gated transduced GFP⁺ cells which

were subsequently intracellularly stained for insulin and glucagon which allowed us to collect a pure population of beta or alpha cells respectively. The stringency of our sorting scheme was also confirmed by qRT-PCR prior to RNA-seq (Supplementary Figure 4c).

4. *What is the GFP+, Insulin+ and Glucagon+ (%) populations in FACS dot plots? Please add this information. Addition of Live/Dead gating would also help. Why there is more alpha cells than beta cells (Fig. 4c) in those FACS plots is not clear. Authors should address this.*

We thank the reviewer for these suggestions. The new revised Figure 4c contains an additional representative FACS plot which shows the Live/Dead gating scheme as requested. We have also included the percentage of cells included within each gate for all three representative FACS plots. The presence of more alpha cells in relation to beta cells is due to different intracellular sorting efficiencies between Insulin⁺ beta cells and Glucagon⁺ alpha cells. As all RNA-seq libraries were constructed from the beta cell fraction we are confident that this variation in sorting efficiency does not affect the interpretation of our data.

5. *To achieve beta cell specific BCL11A deletion mouse model using RIP-Cre is concerning as it has been previously shown that RIP-Cre is expressed both in beta cells and hypothalamus.*

We thank the reviewer for raising this important point. We have successfully used this RIP-Cre mouse line previously to inactivate the calcineurin phosphatase regulatory subunit, calcineurin b1 (Cnb1) in mouse beta cells (Heit JJ, Kim SK, Nature 2006).

We have previously assessed the specificity of the RIP-Cre mouse line in relation to the POMC-Cre mouse line which has activity specifically in the mouse hypothalamus and hippocampus. In the attached figure (page 6 of this document) we assessed brain coronal sections from (a) POMC-Cre; ROSA26R or (b) RIP-Cre; ROSA26R mice. β -galactosidase expression from the ROSA26R reporter gene occurs in the hypothalamus of mice with POMC-Cre (a, blue cells marked by arrowhead), but not in beta cells (c). β -galactosidase expression from the ROSA26 reporter is completely absent in the brains of mice with RIP-Cre (b), but is detected in the beta cells (blue cells, d).

We are confident that the RIP-Cre line used has specific activity in mouse beta cells and hence results in the deletion of Bcl11a specifically in the beta cells of Bcl11a β KO mice. If the editor considers this data critical to the interpretation of the manuscript we can include it as an additional supplementary figure.

6. *Indeed, normal GTT with increased circulating insulin levels in KO mice is indicating an insulin resistance phenotype possible due to the hypothalamic deletion of this gene. To assess whether mouse deletion of BCL11A affects the first phase insulin secretion addition of GSIS with 2-5 min insulin detection might be helpful.*

We thank the reviewer for this important suggestion. To assess the effect of beta cell specific Bcl11a deletion on first phase insulin secretion we carried out insulin perfusion assays on islets isolated from eight week old Control and Bcl11a β KO mice (Supp Figure 5g). Our perfusion assays are capable of detecting changes in secreted insulin at 3 minute intervals in response to changes in glucose or secretagogue concentrations. Our data indicate that Bcl11a β KO islets have an enhanced first phase insulin secretion in

response to 16.7 mM glucose. Additionally, Bcl11a β KO islets secreted significantly more insulin in the presence of IBMX (an artificial beta cell depolarizing agent). Together the data from our new perfusion assays confirm that the knockdown of Bcl11a in mouse beta cells enhances first phase insulin secretion.

7. In addition, to test whether some of the indicated insulin exocytosis genes have similarly altered expression in the mouse model should be examined.

We thank the reviewer for this suggestion. Our RNA-seq analysis in human pseudo islets indicated that BCL11A knockdown resulted in a significant increase in GLUT2 expression. As suggested we assessed the expression of GLUT2 in age-matched Control and Bcl11a β KO pancreatic sections (new Supplementary Figure 6). We observed a two fold increase of GLUT expression in Bcl11a β KO beta cells suggesting that the downstream targets of BCL11A which regulate insulin secretion are likely conserved between human and mouse beta cells.

8. The quality of the IF staining needs to be improved and a percentage of the deletion in beta cells should be reported.

We thank the reviewer for this suggestion. We have included new high resolution images of control and Bcl11a β KO islets stained with Bcl11a (white) and Insulin (red) in revised Supplementary Figure 5b. The percentage of Bcl11a deletion in beta cells has also been quantified and is presented in revised Supplementary Figure 5c.

Minor:

1. Figure 3: The labeling of (c) and (d) is in reverse order in the figure legend.

We thank the reviewer for pointing this out and have made the appropriate corrections in the figure legend.

2. Figure 5b: The y axis label is missing.

We regret this oversight and have included the appropriate axis label in revised Figure 5b.

beta cell specific activity of the RIP-Cre mouse line

Reviewer #1 (Remarks to the Author):

The authors have addressed the issues raised in my original review.

Response to Referees' Comments

Reviewer #1 (Remarks to the Author):

The authors have addressed the issues raised in my original review.

We thank the reviewer for this positive assessment.

Reviewer #2:

Conveyed to the Editorial team that the revision was satisfactory.

We thank the reviewer for his/her positive comments.